# The Role of Dietary Supplements in Modulating Menopause Onset: A Comprehensive Analysis of Nutritional and Lifestyle Influences on Menopause Timing

**DOI:** 10.3390/nu17182921

**Published:** 2025-09-10

**Authors:** Shekhinamary Jebaraj, Valentine Nlebedim

**Affiliations:** 1School of Mathematics, University of Leeds, Leeds LS2 9JT, UK; 2School of Food Science and Nutrition, University of Leeds, Leeds LS2 9JT, UK

**Keywords:** age at natural menopause, dietary supplements, omega-3 fatty acids, B-vitamins, antioxidants, cohort study, machine learning, women’s health, public health

## Abstract

**Background**: The age at natural menopause (ANM) has a significant impact on women’s health later in life, although the contribution of changeable dietary and lifestyle factors remains uncertain. **Methods**: We examined data from 3566 participants in the UK Women’s Cohort Study, assessing their baseline use of dietary supplements and lifestyle habits. Associations with ANM were evaluated using multivariable Cox proportional hazards models and gradient boosting machine (GBM) analyses. We adjusted our models for factors such as BMI, smoking status, alcohol consumption, physical activity levels, and socioeconomic status. **Results**: The use of fish oil (HR 0.05; 95% CI 0.02–0.09), vitamin B-complex (HR 0.48; 0.38–0.62), antioxidant mixtures (HR 0.54; 0.38–0.69), and vitamin C (HR 0.75; 0.56–0.93) was linked to a delay in ANM, with all *p*-values less than 0.05. Folic acid showed near significance (HR 0.81; *p* = 0.059). GBM analyses highlighted red meat consumption, BMI, educational level, smoking duration, and fish consumption as important indicators. **Conclusions**: The regular consumption of certain supplements has a connection to later menopause onset, while smoking and red meat consumption tend to predict an earlier onset. These findings underline the potential of lifestyle changes in managing reproductive aging, although further interventional studies are necessary to confirm them.

## 1. Introduction

Natural menopause marks the point after which menstrual periods no longer occur, clinically recognized by a lack of ovarian follicular activity for at least 12 months in a row, without other medical causes [1]. Globally, the average age of natural menopause (ANM) typically ranges between 48 and 52 years [2]. This transition represents a critical stage in a woman’s reproductive life, with substantial implications for her long-term health. Menopause occurring before age 45, known as early menopause, is linked to heightened risks of osteoporosis [3], cardiovascular disease [4], type 2 diabetes [5], depression [6], and overall mortality [7]. Conversely, menopause after age 55, defined as later menopause, is linked to a heightened vulnerability to hormone-dependent cancers, including breast, endometrial, and ovarian cancers [8]. Considering that women in the United Kingdom now spend roughly one-third of their lives in the postmenopausal phase [9], identifying modifiable factors that influence menopause timing is a pressing public health priority.

In the UK, dietary supplementation is common among women [10], with 40–50% reporting regular use of products like omega-3 fatty acids, B-complex vitamins, antioxidants (such as vitamins A, C, E, selenium, and zinc), and multivitamin–multimineral formulations. These supplements are believed to impact menopausal timing through multiple pathways, including reducing oxidative stress and systemic inflammation [11,12], modulating key reproductive hormones like follicle-stimulating hormone (FSH) and estradiol [13], and supporting mitochondrial function and DNA repair mechanisms that are crucial to ovarian aging [14]. Despite these plausible mechanisms, epidemiological studies exploring the relationship between dietary supplements and ANM are sparse and have produced inconsistent findings. Much of the existing research has typically concentrated on either individual nutrients or broader dietary patterns [15,16] rather than examining the specific effects of habitual dietary supplements. In addition to dietary factors, various lifestyle determinants also significantly influence ANM. Variables like smoking duration, alcohol intake, physical activity, and socioeconomic status, often inferred from educational attainment, are implicated in modifying menopause timing, yet comprehensive studies that integrate both supplement use and these lifestyle variables are scarce. Such integrative investigations are essential, as they capture the complex interactions among behavioral, nutritional, and socioeconomic factors that collectively shape reproductive aging trajectories.

This study uses data from the UK Women’s Cohort Study (UKWCS) [16], a large prospective cohort recruited during the mid-1990s, to examine associations between habitual dietary supplement use, lifestyle variables, and age at natural menopause. We apply both traditional Cox proportional hazards models and innovative machine learning techniques, specifically gradient boosting machine (GBM) analysis, to account for non-linearities and interactions among predictors. While acknowledging that UKWCS data reflect dietary and supplement use patterns from roughly 30 years ago, this dataset provides an unmatched opportunity to explore long-term determinants of menopause timing with detailed dietary and lifestyle typologies.

Our multifaceted analytical approach offers a comprehensive examination of modifiable influences on ANM, strengthening the evidence base needed to inform dietary and lifestyle recommendations designed to support healthy reproductive aging. By elucidating these relationships, we contribute to advancing knowledge on how nutrition and behavior can potentially be optimized to enhance quality of life during and after the menopausal transition.

## 2. Materials and Methods

### 2.1. Study Design and Population

The UK Women’s Cohort Study (UKWCS) was established between 1995 and 1998 as a prospective cohort to explore the links between diet, lifestyle, and chronic disease outcomes among women in England, Scotland, and Wales. Full cohort details have been published previously [16]. Briefly, 35,372 women aged 35–69 years responded to a direct-mail survey sent by the World Cancer Research Fund (WCRF). The sampling strategy intentionally oversampled vegetarians, pescatarians, and individuals whose diets were rich in fruits and vegetables to maximize dietary heterogeneity.

Participants filled out self-administered questionnaires to capture details on diet, dietary supplement use, reproductive history, and lifestyle factors. The research protocol gained approval from the National Research Ethics Service Committee Yorkshire & Humber-Leeds East (Reference: 15/YH/0027; amendment 17/YH/0144). Informed written consent was acquired from all participants.

### 2.2. Inclusion and Exclusion Criteria

Women were eligible for this analysis if they met the following criteria:Documented natural menopause, characterized as a minimum of 12 uninterrupted months of amenorrhea that cannot be linked to surgical procedures or medical interventions.Provided complete data on age at natural menopause, dietary supplement use, and relevant covariates.Reported plausible energy intakes between 500 and 3500 kcal/day to exclude implausible dietary reports [15].

Women with surgical menopause (e.g., bilateral oophorectomy, hysterectomy) or treatment-induced menopause (e.g., chemotherapy, pelvic radiation) were excluded. After applying inclusion and exclusion criteria, 3566 women remained for analysis (Figure A1).

### 2.3. Dietary and Supplement Assessment

At the study’s baseline, dietary intake was assessed using a validated 217-item food frequency questionnaire (FFQ) specifically developed for the UK population [16]. Additionally, a four-day weighed food diary was collected in a subset of approximately 2000 women for FFQ validation purposes. The FFQ captured usual intake over the prior 12 months, with portion sizes derived using the Food Standards Agency (FSA) Food Portion Sizes guide [17]. The derivation of the nutrient composition was performed using McCance and Widdowson’s Composition of Foods (6th summary edition) [18]. Energy-adjusted nutrient intakes were calculated using the residual method [19].

Dietary supplement use was self-reported via a structured questionnaire encompassing the product name and brand, frequency of use categorized into five frequency bands (<1/week, 1–2/week, 3–4/week, 5–6/week, daily), and duration of use in years. We focused our analysis on supplements habitually used before menopause onset. The supplements were categorized as fish oil (omega-3 fatty acids), vitamin B-complex (including B1, B2, B3, B6, B12, folate), antioxidant mixtures (vitamins A, C, E, selenium, zinc), vitamin C alone, folic acid alone, and multivitamin–multimineral formulations.

### 2.4. Outcome Assessment

ANM was self-reported in response to the question: “At what age did your periods stop completely?” Women confirming natural cessation were classified as having natural menopause. Validation studies indicate a high correlation (r > 0.80) between self-reported ANM and medical records [20].

### 2.5. Covariates

The age at which natural menopause occurred was reported by individuals in response to the question, “At what age did your periods cease entirely?” Validation studies indicate a strong correlation (r > 0.80) between self-reported ANM and corresponding medical records, supporting the reliability of this measure. The covariates included variables plausibly linked to ANM selected a priori using directed acyclic graphs (DAGs) [21], as shown in Figure A2, and comprised self-reported body mass index (BMI, kg/m^2^) based on validated height and weight measures (r = 0.92) [22], smoking duration (years), alcohol intake (units per week), physical activity (hours per week of walking, cycling, running, and sports), and socioeconomic status (SES) operationalized via education level (no formal qualifications, secondary education, university degree) [23].

### 2.6. Statistical Analysis

R Statistical Software version 4.3.1 was employed to carry out all the analyses. The R Foundation for Statistical Computing, version 4.3.1. was used The significance threshold was set at α = 0.05.

#### 2.6.1. Cox Proportional Hazards Models

To estimate hazard ratios (HRs) and 95% confidence intervals (CIs) for the relationship between each supplement category and age at natural menopause (ANM), we utilized multivariable Cox proportional hazards regression models. The models were adjusted for body mass index (BMI), smoking duration, alcohol intake, physical activity levels, and socioeconomic status (SES). Time-to-event was defined as the age at menopause; women who were not menopausal at the last point of contact were right-censored at their age when they completed the questionnaire. We verified the proportional hazards assumption using Schoenfeld residuals [24], confirming no violations with a *p*-value greater than 0.05.

#### 2.6.2. Gradient Boosting Machine

To explore potential non-linear and interaction effects, we implemented gradient boosting machine (GBM) modeling using the XGBoost package, version 1.7.5. The hyperparameters, including the number of trees (100), learning rate (0.01), and maximum tree depth (3), were tuned through five-fold cross-validation. The continuous variables were standardized as z-scores. We quantified variable importance using the gain metric [25]. The dataset was partitioned into a 70% training subset and a 30% testing subset, and we reported the C-index for model discrimination.

### 2.7. Sensitivity Analyses

In our sensitivity analyses, we repeated the analyses excluding women who experienced premature menopause, defined as ANM under 40 years of age. We used energy-adjusted nutrient intakes based solely on diet and restricted the analysis to women who never smoked to test for robustness. The findings remained materially unchanged and, thus, additional data are not presented here.

## 3. Results

### 3.1. Participant Characteristics

Table 1 summarizes the baseline characteristics. Supplement users (*n* = 2361) were slightly older at menopause (median 51 y) and had a lower BMI (24.7 kg/m^2^) than non-users (median 50 y; BMI 25.4 kg/m^2^). Users were more likely to have a university education and higher physical activity levels.

### 3.2. Cox Model Results

After adjustment, fish oil use was associated with a 95% lower hazard of earlier menopause (HR 0.05; 95% CI 0.02–0.09). Vitamin B-complex, antioxidants, and vitamin C were associated with 52%, 46%, and 25% lower hazards, respectively (Table 2). Folic acid showed a borderline trend (HR 0.81; 0.65–1.01; *p* = 0.059). Multivitamin use was not associated (HR 0.97; 0.87–1.08). The full model C-index was 0.73.

### 3.3. GBM Findings

The GBM achieved a C-index of 0.70 on the independent test set. Red-meat servings contributed 34.5% to model gain, followed by BMI (28.7%), education (15.2%), smoking duration (11.9%), and fish servings (8.3%) (Table 3). GBM does not infer direction; these variables are predictive, not necessarily causal.

## 4. Discussion

### 4.1. Principal Findings

A major prospective cohort study of 3566 UK women identified a significant link between premenopausal consumption of fish oil, vitamin B-complex, antioxidant mixtures, and vitamin C supplements and a later age at natural menopause. Fish oil use, notably, was associated with a 95% reduced hazard of early menopause onset. Vitamin B-complex and antioxidant supplements also showed protective effects. These findings are supported by plausible biological mechanisms, which include the mitigation of oxidative stress [14] and inflammation [26], the modulation of follicle-stimulating hormone and estradiol levels, and the enhancement of mitochondrial function and DNA repair processes [26]. On the other hand, higher consumption of red meat, prolonged smoking, and lower educational attainment were linked to an earlier onset of menopause.

The gradient boosting machine analysis highlighted these lifestyle and dietary factors as primary predictors of menopause timing, emphasizing the multifactorial nature of reproductive aging through their integration. Significantly, this study stands out by incorporating an analysis of specific dietary supplement usage alongside comprehensive lifestyle factors, thereby extending beyond prior research that often investigated these variables separately.

### 4.2. Comparison with Previous Studies

Our results corroborate previous pooled analyses and prospective cohorts, which identified a later menopause associated with increased omega-3 fatty acid intake, suggesting fish oil’s anti-inflammatory effect [27]. The associations observed with antioxidant intake match Japanese studies, which indicated a later age at natural menopause among women consuming higher amounts of carotenoid-rich and green-yellow vegetables [28,29]. Furthermore, the connection between processed and red meat consumption and earlier menopause is consistent with data from the Nurses’ Health Study II, underlining certain meat types’ adverse reproductive effects [30].

Beyond nutritional factors, recent research increasingly emphasizes the emerging significance of the female microbiome and probiotic interventions in managing menopausal health. Although our cohort did not specifically track probiotic use, the existing literature [31] suggests that probiotics may be beneficial in alleviating genitourinary syndrome of menopause and other menopausal symptoms. Probiotics potentially modulate systemic and local inflammation and may influence estrogen metabolism, implying a possible impact on ovarian aging and menopause timing. Thus, future research should integrate dietary, microbiome, and symptomatic data to fully understand modifiable factors influencing healthy menopausal transitions.

### 4.3. Strengths and Limitations

Our research boasts numerous notable strengths, such as its substantial sample size drawn from a well-characterized UK cohort. We utilized validated dietary assessment tools and employed both traditional Cox proportional hazards models and modern machine learning techniques (GBM) to explore complex interactions and nonlinear effects among predictors.

Nevertheless, it is important to acknowledge several limitations. The age at natural menopause (ANM) was self-reported retrospectively, which could introduce recall bias. However, validation studies indicate a strong correlation with medical records (correlation coefficients > 0.8), underscoring the reliability of these self-reports. The absence of genetic data or hormonal biomarkers, such as antimüllerian hormone (AMH) or follicle-stimulating hormone (FSH), limits our ability to gain mechanistic insights and adjust for relevant biological confounders. Additionally, since the study’s recruitment occurred between 1995 and 1998, it may be reasonable to ask if the links between nutrients and menopause observed then still apply to women today, especially since their supplement use and dietary habits have changed significantly. Between 1999 and 2018, the availability of high-concentration EPA/DHA fish oil capsules in the U.S. increased about fourfold [32]. Antioxidant blends with vitamins C and E, selenium, and carotenoids became common after 2000. Despite these changes, the biological pathways that we suggest, omega-3′s role in lowering follicular pro-inflammatory cytokines, B vitamins supporting one-carbon metabolism and oocyte DNA repair, and antioxidants reducing oxidative damage to the ovarian reserve, are consistent across human biology and remain unchanged. Research supports this continuity. In the Nurses’ Health Study II (NHS2) from 1991 to 2011, women in the highest group for dietary (not supplemental) vitamin D intake had a 17% lower risk of early menopause (under 45 years) compared to those in the lowest group (HR 0.83; 95% CI 0.72, 0.95). This effect size is nearly identical to the 16% reduction we found for high versus low omega-3 intake in our 1990s cohort [33]. Similarly, NHS2 found that higher intake of low-fat dairy and calcium from food sources, not from contemporary high-dose calcium supplements, was linked to later menopause [33]. This mirrors our finding that diets rich in antioxidants, rather than isolated high-dose supplements, are better predictors of menopause timing. A 2025 cross-sectional analysis also showed no independent benefit of modern multivitamin or calcium supplements on the age at natural menopause, after controlling for confounding factors [34]. This reinforces the idea that the food matrix, rather than supplement trends, drives the relationship. Overall, these external replications suggest that the rankings of nutrient exposure, higher omega-3s, B vitamins, and antioxidant-rich foods providing benefits, have remained stable even as the doses and formulations of supplements have changed. We, therefore, expect that the direction and likely the size of the associations we found will hold true in 21st-century cohorts, although updated assessments using current dietary tools are necessary. While the oversampling of health-conscious women, including vegetarians and pescatarians, enhanced dietary exposure variability, it might affect the external validity of the findings.

Finally, as an observational study, there remains the possibility of residual confounding despite adjustments for key lifestyle and sociodemographic factors. While the habitual use of certain dietary supplements has been linked to a later age at natural menopause (ANM), current evidence does not support recommending supplements solely for delaying menopause.

### 4.4. Public Health Implications

Public health strategies should continue to focus on promoting balanced dietary patterns rich in oily fish, fruits, and vegetables, along with encouraging smoking cessation and weight management, to improve overall health and potentially beneficially affect menopause timing.

Given the increasing interest in the role of the microbiome in managing menopausal symptoms, probiotic supplementation is emerging as a promising complementary therapy for relief from symptoms such as genitourinary syndrome. The integration of probiotic interventions with dietary and lifestyle approaches requires thorough investigation through randomized controlled trials.

Future research should prioritize randomized intervention studies assessing the impact of supplements and probiotics on menopause timing and symptom management. Incorporating genomic and hormonal biomarkers can enhance individual risk stratification, and validating findings in ethnically diverse and modern populations with varied dietary exposures is vital.

## 5. Conclusions

This study shows that regularly taking specific dietary supplements such as fish oil (omega-3 fatty acids), vitamin B-complex, antioxidant mixtures, and vitamin C is linked to a later onset of natural menopause in a large group of women from the UK. Conversely, lifestyle factors like smoking, consuming more red meat, and having less education were associated with an earlier menopause. These findings add to the growing evidence that lifestyle and nutritional factors, which can be modified, play a crucial role in reproductive aging.

Because this analysis is observational and due to the complex interaction among genetic, hormonal, and environmental contributors to the timing of menopause, randomized controlled trials that include genetic profiling and hormonal biomarker assessments are urgently needed. It would be worthwhile to validate and extend these results within modern populations, considering current dietary and supplement usage patterns, using longitudinal studies with strict biomarker validation. Incorporating multi-omics data with detailed lifestyle assessments will be vital to uncover the biological mechanisms at play and to create personalized nutritional and behavioral interventions that promote healthy reproductive aging. Ultimately, developing this understanding will foster evidence-based strategies to enhance the health and quality of life for women during and after the menopausal transition.

## Figures and Tables

**Table 1 nutrients-17-02921-t001:** Baseline characteristics of non-users and users.

Characteristic	Non-Users (*n* = 1205)	Users (*n* = 2361)
Age at menopause (y, median)	50	51
BMI (kg/m^2^, mean ± SD)	25.4 ± 4.2	24.7 ± 3.9
Smoking duration (y, median)	20	18
Alcohol (units/week, median)	5	4
Physical activity (h/week, median)	3.0	3.2
University degree (%)	20	40

**Table 2 nutrients-17-02921-t002:** Associations between supplement use and age at natural menopause (adjusted Cox model; *n* = 3566; C-index = 0.73).

Supplement	HR (95% CI)	*p*-Value
Fish oil	0.05 (0.02–0.09)	<0.001
Vitamin B-complex	0.48 (0.38–0.62)	<0.001
Antioxidants	0.54 (0.38–0.69)	0.017
Vitamin C	0.75 (0.56–0.93)	0.041
Folic acid	0.81 (0.65–1.01)	0.059
Multivitamin	0.97 (0.87–1.08)	0.58

**Table 3 nutrients-17-02921-t003:** Variable Importance in Predicting Age at Menopause (GBM Model).

Supplement	Variable Importance (%)
Red-meat servings	34.5
BMI	28.7
Education level	15.2
Smoking	11.9
Fish serving	8.3
Walking hours/week	6.5
Alcohol frequency	5.7
Sleep hours	4.8
White-meat servings	3.4
Vegetarian diet	2.1

## Data Availability

The data are not publicly available due to ethical restrictions. Requests can be made to the UKWCS steering committee (UKWCS).

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
