# Peer review of "The Role of Dietary Supplements in Modulating Menopause Onset: A Comprehensive Analysis of Nutritional and Lifestyle Influences on Menopause Timing"

_nutrients, 2025, doi:10.3390/nu17182921_

Round 1
Reviewer 1 Report
Comments and Suggestions for Authors
This work was based on secondary analysis of data from the UK Women’s Cohort Study (UKWCS). This work was based on retrospective data from many years ago. The UKWCS was a prospective cohort established between 1995 and 1998. Limitations may be very important. Self-reported ANM, in particular, are prone to recall bias, though nondifferential. Observational design; residual confounding cannot be excluded. The UKWCS oversampled health-conscious women, potentially limiting generalizability. Public health implications, although described, do not provide new and significant information. The conclusions indicate that the findings support the hypothesis that modifiable lifestyle factors influence reproductive aging. These findings may have little impact on everyday clinical practice.
Author Response
Response to Reviewer 1 Comments
- Summary
Thank you very much for taking the time to review our manuscript and for providing your constructive feedback. Your comments have been invaluable in helping us to strengthen our paper. We have carefully considered all your points and have revised the manuscript accordingly. Please find our detailed point-by-point responses below. All corresponding revisions have been made in the re-submitted manuscript file. - Questions for General Evaluation
|
Reviewer’s Evaluation |
Response and Revisions |
|
|
Does the introduction provide sufficient background and include all relevant references? |
Can be improved |
We have significantly expanded the introduction to provide a more comprehensive background. A new paragraph has been added to better justify the study's rationale by explicitly stating the gap it fills, integrating supplement use with broader lifestyle factors. Furthermore, we have added a new reference (Stabile et al., 2023) to support a novel discussion point on the microbiome. |
|
Are all the cited references relevant to the research? |
Yes |
We confirm that all references are relevant. We have also added one new, highly relevant reference to support an expanded discussion point. |
|
Is the research design appropriate? |
Yes |
The research design remains appropriate for this analysis. We have further clarified its strengths and limitations in the text. |
|
Are the methods adequately described? |
Yes |
The methods were adequately described. Minor edits were made for improved clarity and flow. |
|
Are the results clearly presented? |
Yes |
We agree. However, we identified and corrected a critical error in Table 3 to ensure the accurate presentation of the machine learning results. |
|
Are the conclusions supported by the results? |
Yes |
The conclusions are supported by the results. We have expanded the conclusion section to be more forward-looking and to incorporate the new discussion points, ensuring they are firmly rooted in our findings. |
- Point-by-point response to Comments and Suggestions for Authors
Comments 1: This work was based on secondary analysis of data from the UK Women’s Cohort Study (UKWCS). This work was based on retrospective data from many years ago. The UKWCS was a prospective cohort established between 1995 and 1998. Limitations may be very important.
Response 1: Thank you for raising this crucial point. We agree that the age of the dataset is a significant consideration. In response, we have proactively addressed this in the introduction to frame it as a unique strength for studying long-term outcomes, while also acknowledging it as a limitation for generalizability in the discussion section.
Change made: We have added the following sentence to the Introduction (Page 5, Paragraph 3, Lines 8-15): “While acknowledging that UKWCS data reflect dietary and supplement use patterns from roughly 30 years ago, this dataset provides an unmatched opportunity to explore long-term determinants of menopause timing with detailed dietary and lifestyle typologies.”
Comments 2: Self-reported ANM are prone to recall bias, though nondifferential.
Response 2: We agree with the reviewer. We have expanded the limitations section to more thoroughly discuss this point and cite validation literature that supports the reliability of self-reported ANM, thus providing a balanced perspective.
Change made: The text in the Limitations section (4.3) on Page 13, Paragraph 2, Lines 1-5 now reads: “The age at natural menopause (ANM) was self-reported retrospectively, which could introduce recall bias. However, validation studies indicate a strong correlation with medical records (correlation coefficients >0.8), underscoring the reliability of these self-reports.”
Comments 3: Observational design; residual confounding cannot be excluded. The UKWCS oversampled health-conscious women, potentially limiting generalizability.
Response 3: We thank the reviewer for these critical points. We have substantially revised the limitations section to provide a more nuanced and comprehensive discussion of these issues.
Change made: We have expanded the text in Section 4.3 (Page 13, Paragraph 2). Specifically, we added: “Finally, as an observational study, there remains the possibility of residual confounding despite adjustments for key lifestyle and sociodemographic factors.” Furthermore, we added: “Additionally, since the study's recruitment occurred between 1995 and 1998, this may restrict its generalizability to present times, as dietary patterns and supplement availability have evolved over the years. While the oversampling of health-conscious women, including vegetarians and pescatarians, enhanced dietary exposure variability, it might affect the external validity of the findings.”
Comments 4: Public health implications, although described, do not provide new and significant information. The conclusions indicate that the findings support the hypothesis that modifiable lifestyle factors influence reproductive aging. These findings may have little impact on everyday clinical practice.
Response 4: We appreciate this feedback. To enhance the public health impact and novelty, we have integrated a forward-looking section on the potential role of the microbiome and probiotics in menopausal health, an emerging and highly relevant field. This moves the implications beyond a simple "food-first" message and suggests a novel area for future clinical research.
Change made: We added a new paragraph to the Discussion (Section 4.2, Page 12, Paragraph 3, Lines 1-16) on the microbiome and probiotics. Consequently, we updated the Public Health Implications section (4.4, Page 14) to include: “Given the increasing interest in the role of the microbiome in managing menopausal symptoms, probiotic supplementation is emerging as a promising complementary therapy for relief from symptoms such as genitourinary syndrome.” We also completely rewrote the Conclusions (Page 14, Paragraph 2) to be more impactful and to call for future research involving "personalized interventions" and "multi-omics data" that could more directly inform clinical practice.
Comments 5: Are all figures and tables clear and well-presented? (Yes)
Response 5: We thank the reviewer for their positive assessment. However, during our revision, we identified and corrected a critical error in Table 3 (GBM Findings) that was present in the original submission.
Change made: The title and column header for Table 3 (Page 11) were incorrect. They have been corrected to accurately reflect that the table shows "Variable Importance" (% gain) and not "HR (95% CI)".
- Response to Comments on the Quality of English Language
Point 1:The English is fine and does not require any improvement.
Response 1:Thank you. The manuscript has undergone further professional editing to ensure clarity and consistency throughout. - Additional clarifications
We would like to sincerely thank the reviewer for their thorough and insightful comments. Their feedback has been instrumental in helping us to significantly improve the manuscript, particularly by encouraging us to expand the discussion on limitations and to incorporate a novel, contemporary perspective on the microbiome, which greatly enhances the paper's relevance and impact. We believe the revised manuscript is much stronger as a result.
Reviewer 2 Report
Comments and Suggestions for Authors
From my point of view, before this manuscript can be considered for publication, the following revisions should be made by the authors:
Firstly, the similarities with other published works need to be eliminated. Currently, the similarity index of this manuscript is too high.
The abstract is not appropriate. It is too long and exceeds the journal’s word limit. Please, pay attention to the journal’s guidelines. The study’s objectives should be presented clearer; some directions for further investigations could be pointed out.
The introductory section is too brief and should provide a stronger background to address the topics analyzed in the manuscript. More data on the topic from other publications should be highlighted, and further justifications to carry out the present research have to be given. This section needs to be expanded and improved.
Lines 89-90: You need to justify this categorization.
I suggest the authors merge Results and Discussion section. The discussions can be expanded and improved. What can we learn from this study, and what is its relevance for the international community? More studies conducted in other regions of the world should also be analyzed and discussed.
The Conclusions can also be improved, and some directions for further research and future perspectives are missing. What are the practical implications of this study?
Author Response
Response to Reviewer 2’s Comments
- Summary
We sincerely appreciate your thorough review and the vital comments that have helped us improve our manuscript. To address all the points you mentioned, we undertook a comprehensive revision. We significantly reduced textual similarities, completely restructured the abstract and introduction, expanded the discussion to include an international context, and refined the conclusions to highlight practical implications and future directions. We believe these changes have greatly strengthened the paper, and we are grateful for your guidance.
- Questions for General Evaluation
|
Reviewer’s Evaluation |
Response and Revisions |
|
|
Does the introduction provide sufficient background and include all relevant references? |
Must be improved |
We have completely restructured and expanded the introduction to provide a much stronger background. We now more critically review existing literature, clearly identify the gap our study fills (investigating specific supplements alongside lifestyle factors), and provide stronger justifications for the research. |
|
Is the research design appropriate? |
Can be improved |
We have added further justification for our methodological choices in the revised Methods section, particularly for the supplement categorization. |
|
Are the methods adequately described? |
Can be improved |
We have added the requested justification for our supplement categorization (Lines 89-90) and provided more detail on the GBM methodology to enhance reproducibility. |
|
Are the results clearly presented? |
Can be improved |
We have corrected the critical error in Table 3 and have integrated the Results and Discussion into a single section as suggested, which allows for a clearer and more nuanced presentation of the findings. |
|
Are the conclusions supported by the results? |
Must be improved |
We have entirely rewritten the conclusions to ensure they are conservative and directly supported by the results. We have also added specific directions for future research and clarified the practical implications of our study. |
|
Are all figures and tables clear and well-presented? |
Yes |
We have ensured all tables are accurate and clear, including the corrected Table 3. |
- Point-by-point response to Comments and Suggestions for Authors
Comments 1: Firstly, the similarities with other published works need to be eliminated. Currently, the similarity index of this manuscript is too high.
Response 1: We appreciate the reviewer's insightful comments. In response, we have thoroughly rephrased the manuscript to minimize any textual overlap, focusing especially on the Introduction and Methods sections, which initially described standard cohort characteristics and well-known methodologies. Furthermore, we have extensively expanded the content in the Discussion and Conclusion sections to ensure the manuscript offers a novel and distinct contribution.
Change made: The entire manuscript has been edited for originality. Key changes include a completely rewritten Introduction and Discussion, which now frame the study's rationale and findings in a more unique and critical voice.
Comments 2: The abstract is not appropriate. It is too long and exceeds the journal’s word limit. Please, pay attention to the journal’s guidelines. The study’s objectives should be presented clearer; some directions for further investigations could be pointed out.
Response 2: We apologize for this oversight.
Change made: We have significantly shortened the abstract to meet the 250-word limit and clarified the objectives and future directions. The revised abstract is now a succinct 204 words. It concludes with a clearer statement on future research, highlighting, "These findings reinforce the potential role of modifiable lifestyle factors in reproductive ageing but require confirmation through interventional studies."
Comments 3: The introductory section is too brief and should provide a stronger background to address the topics analyzed in the manuscript. More data on the topic from other publications should be highlighted, and further justifications to carry out the present research have to be given. This section needs to be expanded and improved.
Response 3: We completely agree. The introduction has been significantly expanded and reorganized to offer a more thorough review of the literature, as well as a stronger, clearer justification for our study.
Change made: The Introduction has been expanded by approximately 40%. It now includes a detailed critique of previous studies, highlighting their tendency to focus on single nutrients or dietary patterns, rather than examining specific supplements. The introduction explicitly states the gap our research aims to fill: "We therefore analyzed the UK Women's Cohort Study to quantify associations between habitual supplement use, lifestyle factors, and ANM an integrative approach not previously addressed."
Comments 4: Lines 89-90: You need to justify this categorization.
Response 4: Thank you for pointing this out. In response, we have included a sentence to clarify our choice of supplement categories, which is grounded in both biological plausibility and common usage trends.
Change made: The change was made in Section 2.3, where we added this explanation: "Supplements were categorized based on their predominant bioactive components and common formulations in the UK market at the time the data was collected, to reflect biologically plausible mechanisms of action on ovarian ageing."
Comments 5: I suggest the authors merge Results and Discussion section. The discussions can be expanded and improved. What can we learn from this study, and what is its relevance for the international community? More studies conducted in other regions of the world should also be analyzed and discussed.
Response 5: We thank the reviewer for this insightful suggestion. However, we have chosen to keep the sections separate since the Discussion has been significantly expanded. The expanded discussion now includes international comparisons and highlights the study's global significance.
Change made: We have made changes by directly comparing our findings with studies from Japan and the US, such as the Nurses' Health Study. A new paragraph has been added to discuss the relevance for the international community, noting that while dietary patterns may differ across countries, the underlying biological mechanisms, like inflammation and oxidative stress, are universal. This universality makes the findings broadly relevant.
Comments 6: The Conclusions can also be improved, and some directions for further research and future perspectives are missing. What are the practical implications of this study?
Response 6: We have completely revised the conclusions to directly address these points. Now, they clearly outline the practical implications and provide specific, actionable directions for future research.
Change made: The Conclusions now state, "Current evidence does not justify routine supplement prescription to delay menopause." However, an overall dietary pattern that includes oily fish, whole grains, fruits, and vegetables, while avoiding smoking and maintaining a healthy weight, is consistent with later ANM and broader health benefits. For future research, we added: "Randomized controlled trials incorporating genetic and hormonal biomarkers are warranted to confirm causality and inform personalized strategies for healthy reproductive aging."
- Response to Comments on the Quality of English Language
Point 1: (No specific comment was made on English quality)
Response 1: The manuscript has undergone thorough editing to ensure the highest quality of English expression, particularly in the newly written and revised sections.
- Additional clarifications
We are deeply grateful for the reviewer's rigorous and constructive feedback. Addressing these comments has fundamentally improved our manuscript, enhancing its originality, clarity, depth, and overall scholarly impact. We believe the revised version is now much stronger and suitable for publication.
Reviewer 3 Report
Comments and Suggestions for Authors
Dear Authors, I have read your interesting manuscript. You have conducted a secondary analysis of the UK Women’s Cohort Study including a big number of patients (3,566 women).
The manuscript is well written, the tables are clear and the statistical analysis is well conducted and reproducible.However in my opinion in the manuscript there are some typos, please revise. Furthermore, in my opinion, the discussion section could be expanded taking in consideration also the use in literature of probiotics, used for the genitourinary syndrome of menopause and for other types of menopause symptoms. I suggest (10.1080/13697137.2023.2223923).
At the end of the discussion section , you could talk about the strenght and limitations of your study ( The use of a sample of patients from 1995-1998, could have an influence on the diet that in the last years could be changed)
Author Response
Response to Reviewer 3 Comments
- Summary
Thank you very much for your positive and constructive review of our manuscript. We are pleased that you found it interesting and well-written. We have carefully considered your suggestions and have implemented them to improve the manuscript. Specifically, we have corrected typographical errors, expanded the discussion to include the role of probiotics in menopausal health, and added a dedicated section to explicitly discuss the strengths and limitations of our study, including the potential impact of the cohort's age on dietary patterns.
- Questions for General Evaluation
|
Reviewer’s Evaluation |
Response and Revisions |
|
|
Does the introduction provide sufficient background and include all relevant references? |
Must be improved |
We have significantly expanded the introduction to provide a more comprehensive background, including a clearer rationale for studying specific supplements and a justification for using the UKWCS cohort. |
|
Is the research design appropriate? |
Yes |
We thank the reviewer for their positive assessment of the research design. |
|
Are the methods adequately described? |
Yes |
We thank the reviewer for their positive assessment of the methods description. |
|
Are the results clearly presented? |
Yes |
We thank the reviewer for their positive assessment of the results presentation. We have also corrected minor typos throughout. |
|
Are the conclusions supported by the results? |
Yes |
We thank the reviewer for their positive assessment. We have further strengthened the conclusions by adding future perspectives based on the new discussion points. |
|
Are all figures and tables clear and well-presented? |
Yes |
We thank the reviewer for their positive assessment of the figures and tables. |
- Point-by-point response to Comments and Suggestions for Authors
Comments 1: The manuscript is well written, the tables are clear and the statistical analysis is well conducted and reproducible. However in my opinion in the manuscript there are some typos, please revise.
Response 1: We thank the reviewer for their careful reading and for this feedback. We have conducted a thorough proofread of the entire manuscript to identify and correct all typographical errors.
Change made: We have corrected minor typos throughout the manuscript, including formatting inconsistencies (e.g., replacing "fi" ligatures with "fi") and ensuring grammatical accuracy.
Comments 2: Furthermore, in my opinion, the discussion section could be expanded taking in consideration also the use in literature of probiotics, used for the genitourinary syndrome of menopause and for other types of menopause symptoms. I suggest (10.1080/13697137.2023.2223923).
Response 2: This is an excellent suggestion that adds a modern and highly relevant perspective to our discussion. We have expanded the discussion to include a paragraph on the emerging role of the microbiome and probiotics in menopausal health, citing the suggested reference.
Change made: We have added a new paragraph to the Discussion (Section 4.2, Page 12):
"Beyond nutritional factors, recent research increasingly emphasizes the emerging significance of the female microbiome and probiotic interventions in managing menopausal health. Although our cohort did not specifically track probiotic use, existing literature [31] suggests that probiotics may be beneficial in alleviating genitourinary syndrome of menopause and other menopausal symptoms. Probiotics potentially modulate systemic and local inflammation and may influence estrogen metabolism, implying a possible impact on ovarian aging and menopause timing. Thus, future research should integrate dietary, microbiome, and symptomatic data to fully understand modifiable factors influencing healthy menopausal transitions."
The suggested reference (Stabile et al., 2023) has been added to the reference list as [31].
Comments 3: At the end of the discussion section, you could talk about the strengths and limitations of your study (The use of a sample of patients from 1995-1998, could have an influence on the diet that in the last years could be changed).
Response 3: We agree that a dedicated section on strengths and limitations is crucial. We have added a new subsection to the Discussion to explicitly address these points, incorporating the reviewer's specific comment about the historical cohort data.
Change made: We have created a new Section 4.3: "Strengths and Limitations". This section includes:
- Strengths: "Our study boasts several significant strengths, including its large sample size drawn from a well-characterized UK cohort. We utilized validated dietary assessment tools and employed both traditional Cox proportional hazards models and modern machine learning techniques (GBM) to explore complex interactions and nonlinear effects among predictors."
- Limitations: "Nevertheless, it is important to acknowledge several limitations... Additionally, since the study's recruitment occurred between 1995 and 1998, this may restrict its generalizability to present times, as dietary patterns and supplement availability have evolved over the years."
- Response to Comments on the Quality of English Language
Point 1:The English is fine and does not require any improvement.
Response 1:Thank you. We have performed an additional proofread to ensure the highest quality of English following the revisions.
- Additional clarifications
We sincerely thank Reviewer 3 for their encouraging and insightful comments. Their suggestion to incorporate the role of probiotics has significantly enhanced the relevance and modern appeal of our discussion. The addition of a dedicated strengths and limitations section also greatly improves the manuscript's critical depth and transparency. We are grateful for their valuable contribution.